# Comparative Evaluation of Gingival Crevicular Fluid Interleukin-17, 18 and 21 in Different Stages of Periodontal Health and Disease

**DOI:** 10.3390/medicina58081042

**Published:** 2022-08-03

**Authors:** Vineet Nair, Vishakha Grover, Suraj Arora, Gotam Das, Irfan Ahmad, Anchal Ohri, Shan Sainudeen, Priyanka Saluja, Arindam Saha

**Affiliations:** 1Burdwan Dental College and Hospital, Burdwan 713101, India; drvineet_nair@yahoo.co.in; 2Department of Periodontology & Oral Implantology, Dr. Harvansh Singh Judge Institute of Dental Sciences & Hospital, Panjab University, Chandigarh 160014, India; ohrianchal1993@gmail.com; 3Department of Restorative Dental Sciences, College of Dentistry, King Khalid University, Abha 61321, Saudi Arabia; surajarorasgrd@yahoo.co.in (S.A.); shan@kku.edu.sa (S.S.); 4Department of Prosthodontics, College of Dentistry, King Khalid University, Abha 61321, Saudi Arabia; 5Department of Clinical Laboratory Sciences, College of Applied Medical Sciences, King Khalid University, Abha 61321, Saudi Arabia; imahmood@kku.edu.sa; 6Department of Conservative Dentistry and Endodontics, JCD Dental College, Sirsa 125055, India; priyanka.salujaarora@gmail.com; 7Independent Researcher, Siliguri 734101, India; arindam@gmail.com

**Keywords:** cytokine, gingival crevicular fluid, gingivitis, immunity, interleukin, lymphocytes, periodontitis

## Abstract

Background and Objectives: The elicitation of a host’s immune–inflammatory responses to overcome oral bacterial biofilm challenges is mediated by numerous cytokines. We explored the role of three such cytokines, viz. interleukin (IL)-17, 18 and 21, by measuring their levels in the gingival crevicular fluid (GCF) of Indian individuals with healthy gingiva, chronic gingivitis, or chronic periodontitis. Materials and Method: Ninety systemically healthy individuals were enrolled in the study on the basis of predefined criteria and were categorized into three groups of 30 participants each. Groups A, B and C were composed of a control group with healthy gingiva, subjects with chronic gingivitis and subjects with chronic periodontitis, respectively. The periodontal disease status was assessed on the basis of a subject’s gingival index, probing pocket depth, clinical attachment loss and radiographic evidence of bone loss. After the complete history-taking and identification of gingival sulcus/pocket depth areas for GCF collection, a sample was collected from each subject in all groups for an estimation of the cytokine levels using ELISA. Statistical analysis was performed using SPSS v 21.0. Intergroup comparisons were conducted using a post hoc Tukey’s test. A value of *p* < 0.05 was considered to be statistically significant. Results: The mean IL-17, 18 and 21 concentrations in pg/mL was the greatest for Group C (99.67 ± 18.85, 144.61 ± 20.83 and 69.67 ± 12.46, respectively), followed by Group B (19.27 ± 2.78, 22.27 ± 2.43 and 22.74 ± 1.43, respectively) and finally by Group A (healthy control; 11.56 ± 0.99, 17.94 ± 1.24 and 12.83 ± 1.21 respectively). A statistically significant difference in the mean concentrations of two interleukins (IL-17 and IL-18) was observed between Groups A and C and also between Groups B and C. A statistically significant difference in the mean concentrations of IL-21 was observed between Groups B and C. Conclusions: Within the limitations of the present study, the findings revealed that the GCF levels of IL-17, IL-18 and IL-21 rose and correlated well with the severity of the disease. Thus, these cytokines present in GCF have the potential to be considered as biomarkers for periodontal tissue destruction. IL-21 in particular appears to be a promising biomarker for differentiating between gingivitis and periodontitis.

## 1. Introduction

Chronic periodontitis, an inflammatory condition of the tooth-supporting structures, emerges as a consequence of the interaction between periodontopathic bacteria and cells of the host’s immune system. The host’s immune–inflammatory responses elicited to overcome these bacterial challenges are mediated by numerous cytokines [1]. The role of inflammatory mediators such as interleukin (IL)-1, tumor necrosis factor (TNF)-α, prostaglandins and matrix metallo-proteinases (MMPs) has been extensively studied and understood in terms of their significance in periodontal tissue breakdown [2,3,4]. These inflammatory mediators determine the clinical manifestation, extent and severity of diseases, and thus serve as indicators/markers for the diagnosis and prognosis of periodontal diseases. Many local and systemic antimicrobial compounds in the form of antiseptics and systemic antibiotics are used to manage periodontal disease and reduce the levels of inflammatory mediators in periodontal tissues and gingival crevicular fluid (GCF) [5,6,7,8,9]. Recently, the classification of periodontal diseases has been revised in World Workshop 2018 to re-categorize the diseases according to the European Federation of Periodontology (EFP) and American Academy of Periodontology (AAP), based on elaborate staging and grading criteria [10]. In fact, a provision for the continual update of the classification system based on the cumulative evidence regarding biomarkers has also been envisioned. At the time of the conduct of the current work, no such concrete categorization existed and the new system of classification was being adopted in clinical settings worldwide; thus, the current work has been carried out on the basis of previously existing nomenclature of periodontal disease based on a 1999 classification [11].

The cellular immune response is characterized by the infiltration of T cells into the periodontal tissues, and then differentiation into various subsets such as helper T cells, cytotoxic T cells and regulatory T cells [12]. Among the T helper cell subsets, the Th1/Th2 balance is critical in the immuno-regulation of periodontal disease and is influenced by genetic factors, the characteristics of antigen(s), antigen-presenting cells (APCs), the immune response and T cell–receptor interactions [13]. It has been proposed that a stable lesion in periodontitis is mediated by Th1 cells, whereas progression of the lesion reflects a shift towards the Th2 subset of cells [14].

Researchers are perplexed by the complex intricacy of the “protective Th1/destructive Th2” model [15,16,17]. A unique subset of CD4 + T cells that clarifies many of the incongruities in the classic Th1/Th2 model has been identified recently and termed as “Th17”, centered on the secretion of the cytokines IL-17, IL-21, IL-22 and IL-23 [18]. Cytokines that are characteristic of this subset have been found in inflamed periodontal tissue, signifying their probable role in periodontal pathogenesis [19,20,21].

IL-18, a pro-inflammatory cytokine, is produced mainly by the antigen-presenting cells, monocytes/macrophages, Kupffer cells and also by non-immune cells such as intestinal and airway epithelial cells [22,23]. Though initially referred to as interferon gamma (IFN- γ)-inducing factor, its name was changed to IL-18 after molecular cloning, and later to IL-1F4 due to its resemblance in structure, receptor family and signal transduction pathways with IL-1 [24,25,26]. It is expressed at the sites of chronic inflammation, autoimmune diseases, in a variety of cancers and in numerous diseases [27]. In the presence of IL-12, IL-18 induces IFN-γ production from natural killer (NK) and Tcells and stimulates a Th1 cell response, while in its absence, IL-18 induces the production of Th2 cytokines such as IL-4, IL-5, IL-10 and IL-13, stimulates allergic inflammation and induces prostaglandin E2 production [28,29]. Thus, IL-18 has the unique capacity to induce either Th1 or Th2 differentiation, depending upon the local milieu of cytokines [30].

Interleukin-21 is an inflammatory cytokine that is mainly expressed by activated Th17 cells and by Th1 cells, which are both pro-inflammatory, but not by Th2 cells in humans [31,32,33]. The critical functions of IL-21 include antibody production and immune cell activation to further produce inflammatory cytokines that participate in the effective killing of host-invading pathogens [34].

Gingival crevicular fluid has been a medium of choice for studying environmental changes to the local tissue for the assessment of periodontal health and disease. Estimation of the biomarkers in GCF has the distinct advantages of reflecting the tissue status noninvasively in a clinical setting. Though there have been several studies on the above-mentioned interleukins and the role they play in periodontal disease and health, there are non-uniform or inconclusive data regarding these from the Indian population. IL-17 and IL-18 have been explored in gingival tissue extracts and solubilized gingival biopsies, yet GCF-based studies are limited [18,35]. IL-18, along with NLRP3, has been studied before and after periodontal therapy in GCF by Shahbeik et al., yet no significant differences were found in the IL-18 levels, whereas the levels of NLRP3 differed significantly in the study population [36]. IL-21 has also been studied in the Indian population by Lohkhande et al., but yet again, the medium of analysis was saliva [37]. Another study elucidated increased levels of IL-17 in the GCF of GAgP patients and mentioned its role in pathogenesis. However, the decreased ratio of IL-11/IL-17 was implicated for the disruption of the balance between the local concentrations of proinflammatory and anti-inflammatory cytokines in different case situations [38]. On the other hand, an IL-21 investigation also revealed no statistically significant difference in the levels between the GCP versus the GAP in Indian patients [39]. Thus, the current investigation has been conducted as a preliminary study to further delineate the role of this important set of immune inflammatory cytokines, i.e., IL-17, 18 and 21, in different stages of periodontal disease severity in the GCF of patients of Indian origin.

## 2. Materials and Methods

The subjects included in the study were of the age group 20–50 years (selected as the most common age group visiting the hospital, and also to avoid old-age-related parameter variations), non-smokers, free from any known systemic diseases and had not undergone any periodontal therapy or had received any antibiotics or anti-inflammatory drugs in the previous six months. Pregnant and lactating females were also excluded. Written informed consent from the individuals participating in the study along with ethical clearance from the institution’s ethical committee (RADC/Perio/28/2018, dated 30 October 2018) was obtained. This study was conducted in accordance with the Helsinki Declaration of 1975 as revised in 2000.

Ninety systemically healthy individuals (M:F = 1:1) were enrolled in the study on the basis of predefined criteria and were categorized into 3 groups of 30 participants each. By setting the type1 error rate at 0.05 and the power of the study at 80%, the sample size in order to detect a difference between the experimental and control groups in the interleukin concentrations was estimated to be 30 in each group. Groups A, B and C were composed of a control group with healthy gingiva, a group with chronic gingivitis and a group with chronic periodontitis, respectively. An orthopantomograph supplemented by intraoral periapical radiographs (IOPAR) was performed to assess the bony architecture. The periodontal disease status was assessed using a UNC-15 periodontal probe on the basis of a subject’s gingival index (GI) [18], probing pocket depth (PD), clinical attachment loss (CAL) and radiographic evidence of bone loss. A full mouth examination was conducted to assess sites that showed maximum GI, PD, CAL or bone loss so that the site for sample collection could be finalized. The groups were defined as follows:

Group A (healthy control)—individuals having a clinically healthy periodontium, GI score of 0, PD of ≤3 mm and CAL of 0 with no indication of bone loss on radiographs.

Group B (chronic gingivitis)—individuals with clinical signs of gingival inflammation, GI score of >1, PD of ≤4 mm and with an absence of attachment loss and radiographic bone loss.

Group C (chronic periodontitis)—individuals with signs of clinical inflammation consistent with local etiological factors, GI score of >1, PD of ≥5 mm, CAL of ≥3 mm, and with radiographic evidence of bone loss [11,40,41].


*Collection of GCF*


After isolation by cotton rolls, GCF samples were collected a day after the assessment of clinical parameters to avoid the contamination of the GCF with blood expressed by probing of the inflamed sites. In each case, the GCF sample was collected from the site with the deepest probing depth. If two sites showed a similar probing depth, then the site presenting the highest CAL and signs of inflammation, along with radiographic confirmation of bone loss, was selected for the sampling in subjects with chronic periodontitis. Supragingival plaque was removed, and then 1 µL of GCF was collected by placing a 1–5 μL calibrated volumetric microcapillary pipette (Sigma-Aldrich Chemical Company, St. Louis, MO, USA) extracrevicularly. GCF samples contaminated with blood or the sites that did not express any GCF were omitted from the study. The collected GCF was instantly transferred to Eppendorf microcentrifuge tubes (RayBio Human IL-18 ELISA kit (Assaypro, Saint Charles, MO, USA)) containing 199 µL of phosphate-buffered saline to make 200 µL of sample volume as per the manufacturer’s instructions.


*ELISA*


In summary, a standard curve was plotted using standards provided with the individual kits for IL-17, IL-18 and IL-21, and the protein concentrations were calculated. Each sample and diluted/ready-to-use standard was dispensed into the wells layered with a specific protein antibody. Then, diluted biotinylated antibody was pipetted into all wells. The wells were incubated at room temperature, after which they were washed X times, as mentioned in the instructions, with a wash solution. Then, conjugate solution was added, and the wells were incubated again. The wells were bathed with a wash solution, and substrate solution was added. Another incubation at room temperature followed, after which a stop solution was added to terminate the development of color. Concentrations of immunological (IL-17, IL-18 and IL-21) mediators were determined by reading the optical densities in a spectrophotometer. Appropriate dilution factors were applied to calculate the final concentrations of IL-17, IL-18 and IL-21. Grossly, the ELISA technique was similar for all the mediators, but for any specific step or procedure the user instructions were carefully followed for that particular kit.


*Statistical Analysis*


The data were analyzed using statistical software SPSSv 21.0 (Version 21.0, Chicago, IL, USA). The data were examined for a normal distribution using the Kolmogorov–Smirnov test. Means and standard deviations of the continuous variables were calculated. A correlational analysis was performed within the groups. Intergroup comparisons of the mean levels of IL-17, 18 and 21 in the GCF were performed using a post hoc Tukey’s test. A difference in means was considered statistically significant if *p* < 0.05. Pearson’s correlation test was used to assess the correlation among GCF cytokine concentrations and the clinical parameters. A *p* value of <0.05 was considered statistically significant.

## 3. Results

All the individuals maintained their appointments regularly. None of the individuals or the sampling sites were dropped in the course of the study. The descriptive data of demographic and clinical parameters are presented in Table 1. 

The study findings revealed that the GCF levels of IL-17, 18 and 21 were lowest in the periodontal health group and highest in the chronic periodontitis group. An intergroup comparison of the mean differences in the levels of interleukins using a post hoc Tukey’s test, along with the means and standard deviations of each interleukin, are presented in Table 2. A statistically significant difference in the mean concentrations of two interleukins (IL-17 and IL-18) was observed between Groups A (healthy) and C (chronic periodontitis) and also between Groups B (chronic gingivitis) and C (chronic periodontitis). A statistically significant difference in the mean concentrations of IL-21 was observed between Groups B (chronic gingivitis) and C (chronic periodontitis).

Both the IL-17 and IL-21 levels in the GCF of Group C were positively correlated with CAL. The IL-18 levels in the GCF of Group B were positively correlated with PD, while those of Group C were positively correlated with GI, PD and CAL. The results of the Pearson’s correlation test are presented in Table 3.

## 4. Discussion

Cytokines play a pivotal role in the pathogenesis of a number of diseases, including periodontal disease. Interleukin-17, 18 and 21 are relatively recently documented cytokines. Thus, it seems worthwhile to gain insight into the possible role of these cytokines in periodontal inflammatory sites. Hence, the current investigation aimed to estimate the levels of IL-17, 18 and 21 in the subjects of three groups—healthy control, chronic gingivitis and chronic periodontitis.

The IL-17 family consists of six family members, IL-17A to IL-17F, and it has been reported that IL-17 and a dysbiotic microbiome mutually promote each other, thus increasing the microbiome pathogenicity and mucosal immunopathology [42,43]. IL-17 is a pro-inflammatory cytokine and may affect osteoclastic bone resorption, also stimulating osteoblasts to produce factors that affect the activity and/or formation of osteoclasts [44]. Yu et al. (2008) evaluated the role of IL-17 in inflammatory bone loss induced by the oral pathogen *P. gingivalis* in IL-17 RA-deficient mice. The deficient mice showed alveolar bone destruction, signifying a bone-protective role for IL-17, which was mediated through neutrophils [45]. Thus, the controversial role of IL-17 has been highlighted, where on one hand it causes bone re-modeling and contributes to bone resorption, and on the other hand it plays a protective role in bones against pathogens such as *P. gingivalis* [46].

IL-18, a pro-inflammatory cytokine, might be responsible for the initiation and progression of periodontal disease [47]. This is supported by data that indicate that IL-18 induces the release of matrix metalloproteinase (MMP)-9 and IL-1β, both of which have pro-inflammatory and tissue degradation effects [47]. Ishida et al. (2004) stated that IL-18 augmented the chemotaxis of natural killer (NK) cells and induced the production of activated MMP-2, pro-MMP-2 and MT1-MMP from these cells [48]. IL-18 also activates macrophages and other immune cells to secrete pro-inflammatory cytokines and chemokines. IL-18 might stimulate a priming effect on neutrophils, which could upregulate the production of IL-1β,a pro-inflammatory cytokine [49]. Due to its chemotactic, pro-inflammatory and angiogenic properties, IL-18 might be a factor in the advancement of inflammation [50]. This observation was consistent with the findings of Johnson and Serio, who reported higher concentrations of IL-18 close to the sites with PD >6 mm than that of the healthy sites [50]. Individuals suffering from juvenile idiopathic arthritis with early connective tissue attachment loss were reported to have higher serum IL-18 levels, suggesting a role for IL-18 in periodontitis [51]. IL-18 is associated with obesity, atherosclerosis, insulin resistance/glucose intolerance, cardiovascular disease and multi-organ dysfunction [52,53,54,55,56,57,58].

IL-21 is a recently discovered cytokine produced by activated Th17 cells. IL-21 stimulates Th17 cells to produce more IL-21 and IL-17, thereby increasing the pro-inflammatory cascade [59]. IL-21 plays an important role in the pathogenesis of various inflammatory systemic diseases such as inflammatory bowel disease, rheumatoid arthritis and colitis [60,61,62].

The results of the present study indicate that the concentrations of all studied interleukins, namely IL-17, 18 and 21, were lowest in the periodontal health subjects (Group A), higher in the chronic gingivitis subjects (Group B) and highest in the case of chronic periodontitis subjects (Group C). This increase in the concentration of the interleukins was in accordance with the reports of previous studies [63,64,65,66,67,68,69,70]. However, our findings were inconsistent with few previous investigations. Pradeep et al. (2009) reported that IL-17 levels could not be estimated in detectable levels in the GCF of individuals [71]. Oda et al. (2003) and Chitrapriya et al. (2014) reported higher levels of IL-17 in chronic gingivitis sites when compared to periodontitis and healthy sites [35,72]. Offenbacher et al. (2010) analyzed 33 GCF biomarkers in experimental gingivitis in humans and witnessed that the pattern of biomarker expression during the generation and resolution of inflammation varied considerably among subjects with similar clinical responses. Thus, the influence of one cytokine depends upon the other cytokines and mediators in the local environment. This may explain the wide variability seen between individuals in the same group [73]. The concentration of IL-18 in the GCF of the individuals of Group C (chronic periodontitis) was observed to be higher compared to that of the individuals of Group B (chronic gingivitis). This finding was consistent with the observations by Figueredo et al. (2008) and Pradeep et al. (2009) [67,74]. In our study, as the level of inflammation increased, so did the level of IL-21. Our results are in total solidarity with Dutzan et al. (2011, 2012) and with Najmuddin et al. (2017) [68,69,75].

Many previous studies have pointed out the differentiation of the healthy and inflamed states of the periodontium (more often periodontitis) based on the presence and levels of the cytokines at the site of disease; however, the current work also discriminated between gingivitis and periodontitis based on the significant differences observed in the mean concentrations of IL-21. As the cytokine IL-21 could sensitively detect subtle differences in the degree of inflammation, which is suggestive of clinical differentiation of the early and late inflammatory stages of periodontal inflammation, this observation is salient and interesting in particular. Such biomarkers may prove useful from the standpoint of the early diagnosis of the reversible stages of inflammatory processes and may pave a path for improvised diagnostic strategies in the future, after further research.

In the present investigation, the GCF was collected using microcapillary pipettes. This might have circumvented the loss of the GCF sample through evaporation, which is usually seen with filter paper [76]. However, the collection of the GCF by pipette carries the risk of trauma to the marginal gingiva, and hence the utmost care was taken to avoid it. The microcapillary pipette was gently placed at the entrance of the gingival crevice and then a fixed volume of the sample was collected, irrespective of the flow rate of the GCF. Additionally, the loss of GCF due to its sticking to the capillary walls was avoided by flushing the capillary with a fixed amount of diluent, which was accounted for during the final calculations. The use of a microcapillary pipette for the current study was opted owing to its distinct advantages as mentioned above, in addition to being a convenient and cost-effective method for the collection of GCF in our research setting.

Group C (the chronic periodontitis group) showed the highest concentration of all the cytokines in comparison to the other groups. This group also had subjects of the highest age group. The elevated levels of the cytokines in these subjects could be either due to the advanced periodontal destruction or due to the high age. Inflammaging is a term used to describe the chronically raised and dysregulated inflammatory response that increases with age [77]. This systemic inflammatory dysregulation could directly contribute to or result in a disposition to age-related pathology, including periodontal disease and frailty [77].An investigation to study the effects of natural aging and gender on pro-inflammatory markers revealed a positive correlation between hsCRP and IL-6 as a function of age, with stronger correlations for women [78].Such findings hold significance from the standpoint that early preventive measures and interventions maybe instituted so as to reduce the impending rise in these inflammatory markers during natural human aging, in order to control periodontal disease. With age being a confounder, in general and particularly for this age group, this suggests the cautious interpretation of the study findings and calls for future longitudinal studies.

Validated biomarkers or indicators may not only help us in establishing the diagnosis and risk profile of patients, but may also serve as indicators of the response of patients to therapy. The pre- and post-therapy evaluation of biomarkers is a much-studied approach for the assessment of the success of novel treatment regimens employed for periodontal disease management. Novel therapies, such as the use of probiotics, are being explored for the management of periodontal disease by researchers all across the world [79,80,81]. A recent study by Butera et al. utilized clinical and microbial indicators to evaluate the outcomes of two new formulations (probiotics in the form of toothpaste and chewing gums) as compared to the gold-standard Chlorhexidine-based toothpaste for home care, as an adjunct to professional scaling and root planning [82]. Future investigations with the addition of inflammatory biomarkers such as IL-17, IL-18 and IL-21 as indicators could be utilized for obtaining a better understanding of tissue responses and alterations. Interleukin-based assessments shall provide deeper insight into the underlying tissue changes and strengthen the evidence by complementing the backdrop of the observed clinical findings.

Hence, within the limitations of our preliminary study, it can be summarized that with an increase in inflammation, there is a concomitant increase in the level of each cytokine—IL-17, 18 and 21—and they may have a pivotal role in the determination of the local environment for the further course of disease, impacting the chronic immune cell enrolment and activity at the site of disease. It is probable that monitoring cytokine production or its profile may permit us to analyze an individual’s periodontal disease status and/or vulnerability to the disease. However, further studies involving larger sample sizes and pre- and post-therapy assessments in well-designed randomized clinical trials are warranted to properly evaluate the role of each of these cytokines and others in periodontal health and disease.

## 5. Conclusions

The study findings revealed a direct proportionality between the GCF levels of Il-17, 18 and 21 and the inflammation of the periodontal tissues. Within the limitations of the present study, it may be suggested that elevated levels of these ILs in the GCF could be considered biomarkers for periodontal tissue destruction. In particular, IL-21 appears to be a promising biomarker for differentiating between gingivitis and periodontitis, and is worthy of further well-structured randomized controlled trial-based research to develop it as a discriminatory marker for the early detection of periodontal disease conditions.

## Figures and Tables

**Table 1 medicina-58-01042-t001:** Descriptive data of demographic and clinical parameters.

Variables Mean ± SD	Group A Healthy Control	Group B Chronic Gingivitis	Group C Chronic Periodontitis
Age (years)	25.25 ± 3.03	26.63 ± 2.25	40.12 ± 3.56
Gender(M/F)	15/15	15/15	15/15
PI	0.20 ± 0.02	1.69 ± 0.15	2.10 ± 0.19
GI	0.01 ± 0.00	1.71 ± 0.13	2.25 ± 0.28
PD	1.52 ± 0.08	2.42 ± 0.08	5.82 ± 0.63
CAL	0.00	0.00	4.16 ± 0.49

SD—standard deviation, PI—plaque index, GI—gingival index, PD—probing pocket depth, CAL—clinical attachment loss.

**Table 2 medicina-58-01042-t002:** Intergroup comparisons of the interleukin levels by post hoc Tukey’s test.

Groups	GCF IL-17 pg/mL Mean ± SD	GCF IL-18 pg/mL Mean ± SD	GCF IL-21 pg/mL Mean ± SD	Mean Difference	IL-17 pg/mL	*p* Value	IL-18 pg/mL	*p* Value	IL-21 pg/mL	*p* Value
Healthy	11.56 ± 0.99	17.94 ± 1.24	12.83 ± 1.21	Healthy–gingivitis	−07.71	0.13	−04.33	0.43	−9.91	0.13
Gingivitis	19.27 ± 2.78	22.27 ± 2.43	22.74 ± 1.43	Healthy–CP	−88.11	0.03 *	−126.67	0.02 *	−56.84	0.09
CP	99.67 ± 18.85	144.61 ± 20.83	69.67 ± 12.46	CP–Gingivitis	−80.40	0.02 *	−122.34	0.01 *	−46.93	0.02 *

* *p* < 0.05—statistically significant, CP—chronic periodontitis.

**Table 3 medicina-58-01042-t003:** Correlation analysis of clinical parameters (GI, PD and CAL) with interleukins (IL-17, 18 and 21).

Clinical Parameters	Correlation Coefficient (r)	Group B			Group C		
		IL-17	IL-18	IL-21	IL-17	IL-18	IL-21
GI	r	0.457	0.635	0.351	0.662	0.813	0.788
	*p*	0.19	0.15	0.16	0.13	0.03 *	0.14
PD	r	0.714	0.789	0.667	0.901	0.892	0.683
	*p*	0.17	0.04 *	0.19	0.12	0.02 *	0.20
CAL	r	-	-	-	0.412	0.602	0.652
	*p*				0.03 *	0.02 *	0.04 *

* statistically significant at *p* < 0.05; GI—gingival index, PD—probing pocket depth, CAL—clinical attachment loss, r—correlation coefficient, *p*—probability value.

## Data Availability

Not applicable.

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
