# Peer review of "Comparative Evaluation of Gingival Crevicular Fluid Interleukin-17, 18 and 21 in Different Stages of Periodontal Health and Disease"

_medicina, 2022, doi:10.3390/medicina58081042_

Round 1
Reviewer 1 Report
Manuscript of interest for the dental sector, especially for periodontists and dental hygienists.
Before you can proceed with a final assessment you need a review.
Title: write the acronym GCF in full
Keywords: these are few, to add others
Introduction: Add the new classification of periodontal disease.
Add the various effects of antiseptics and antibacterials on the market to reduce the inflammatory indexes of the crevicular fluid.
Materials and methods; To better analyze the inclusion and exclusion criteria, those present are few.
How was the sample size calculated?
Discussion: to be added as future objectives how they affect the use of probiotics, paraprobiotics and post biotics to reduce inflammatory indices in crevicular fluid, already studied in the research group of Prof. Scribante.
Conclusions: Add proactive action
Bibliography: add references required in the introduction and discussion
Author Response
|
Sr. no. |
remark |
Response to remark |
|
Reviewer 1 |
|
|
|
1. |
Manuscript of interest for the dental sector, especially for periodontists and dental hygienists. Before you can proceed with a final assessment you need a review. · Title: write the acronym GCF in full
· Keywords: these are few, to add others
· Introduction: Add the new classification of periodontal disease.
· Add the various effects of antiseptics and antibacterials on the market to reduce the inflammatory indexes of the crevicular fluid.
· Materials and methods; To better analyze the inclusion and exclusion criteria, those present are few.
· How was the sample size calculated?
· Discussion: to be added as future objectives how they affect the use of probiotics, paraprobiotics and post biotics to reduce inflammatory indices in crevicular fluid, already studied in the research group of Prof. Scribante.
· Conclusions: Add proactive action
· Bibliography: add references required in the introduction and discussion
|
Necessary modifications made, as suggested and acronym GCF in full has been provided in the title.
Necessary modifications made, as suggested and two new keywords Immunity; Lymphocytes; have been provided.
Necessary modifications made, as suggested and relevant details have been added in the “introduction” section along with the addition of 2 new references in the bibliography. Caton JG, Armitage G, Berglundh T, Chapple ILC, Jepsen S, Kornman KS, Mealey BL, Papapanou PN, Sanz M, Tonetti MS. A new classification scheme for periodontal and peri-implant diseases and conditions - Introduction and key changes from the 1999 classification. J Clin Periodontol. 2018 Jun;45 Suppl 20:S1-S8. doi: 10.1111/jcpe.12935. PMID: 29926489.
Armitage GC. Development of a classification system for periodontal diseases and conditions. Ann Periodontol. 1999 Dec;4(1):1-6. doi: 10.1902/annals.1999.4.1.1. PMID: 10863370.
Necessary modifications made, as suggested and relevant details have been added in the “introduction” section (line no. ) along with the addition of 5 new references in the bibliography.
· Asbi T, Hussein HA, Horwitz J, Gabay E, Machtei EE, Giladi HZ. A single application of chlorhexidine gel reduces gingival inflammation and interleukin 1-β following one-stage implant placement: A randomized controlled study. Clin Implant Dent Relat Res. 2021 Oct;23(5):726-734. doi: 10.1111/cid.13041. Epub 2021 Aug 11. PMID: 34378862. · Lee MK, Ide M, Coward PY, Wilson RF. Effect of ultrasonic debridement using a chlorhexidine irrigant on circulating levels of lipopolysaccharides and interleukin-6. J Clin Periodontol. 2008 May;35(5):415-9. doi: 10.1111/j.1600-051X.2008.01221.x. Epub 2008 Mar 13. PMID: 18341600. · Kaur H, Grover V, Malhotra R, Gupta M. Evaluation of Curcumin Gel as Adjunct to Scaling & Root Planing in Management of Periodontitis- Randomized Clinical & Biochemical Investigation. Infect Disord Drug Targets. 2019;19(2):171-178. doi: 10.2174/1871526518666180601073422. PMID: 29852877. · Emingil G, Atilla G, Sorsa T, Luoto H, Kirilmaz L, Baylas H. The effect of adjunctive low-dose doxycycline therapy on clinical parameters and gingival crevicular fluid matrix metalloproteinase-8 levels in chronic periodontitis. J Periodontol. 2004 Jan;75(1):106-15. doi: 10.1902/jop.2004.75.1.106. PMID: 15025222. · Gupta S, Chhina S, Arora SA. A systematic review of biomarkers of gingival crevicular fluid: Their predictive role in diagnosis of periodontal disease status. J Oral Biol Craniofac Res. 2018 May-Aug;8(2):98-104. doi: 10.1016/j.jobcr.2018.02.002. Epub 2018 Feb 10. PMID: 29892530; PMCID: PMC5993463.
Necessary modifications made, as suggested and relevant details have been added in the “Materials and methods;” section
Necessary modifications made, as suggested and relevant details have been added in the “ Materials and methods;” section
Necessary modifications made, as suggested and relevant details have been added in the “Discussion;” section (line no. ) along with the addition of 3 new references in the bibliography including the suggested research group,s findings as well.
· Schmitter T, Fiebich BL, Fischer JT, Gajfulin M, Larsson N, Rose T, Goetz MR. Ex vivo anti-inflammatory effects of probiotics for periodontal health. J Oral Microbiol. 2018 Jul 25;10(1):1502027. doi: 10.1080/20002297.2018.1502027. PMID: 30057719; PMCID: PMC6060379. · Butera A, Gallo S, Maiorani C, Molino D, Chiesa A, Preda C, Esposito F, Scribante A. Probiotic Alternative to Chlorhexidine in Periodontal Therapy: Evaluation of Clinical and Microbiological Parameters. Microorganisms. 2020 Dec 29;9(1):69. doi: 10.3390/microorganisms9010069. PMID: 33383903; PMCID: PMC7824624. · Liu J, Liu Z, Huang J, Tao R. Effect of probiotics on gingival inflammation and oral microbiota: A meta-analysis. Oral Dis. 2022 May;28(4):1058-1067. doi: 10.1111/odi.13861. Epub 2021 Apr 29. PMID: 33772970.
Necessary modifications made, as suggested
All new references added in the bibliography list as suggested (from the introduction and discussion section additions)
|

Reviewer 2 Report
Major Comments:
In general, the study is well designed and executed.
As the study is focused on the characterization of IL-17/IL-18/IL-21 concentration during the different stages of periodontal health/disease, I would suggest to show the results highlighting the most relevant/striking findings of the study.
I would suggest that on Table 1 you only show age, sex (please add this parameter) and clinical parameters (PI/GI/PD, etc.) evaluated for each group. Then, the results obtained for the cytokines concentration for each group (main finding of the study) may be presented on a separate figure including the p values shown on table 2. The p values obtained by tukey´s posthoc test can be incorporated into the figure and/or showed on as a supplementary table, as they are not a result itself, but a statistical analysis.
Along the cohort of patients, group C has the highest mean age. I understand that chronic periodontitis is more prevalent on older people, but the upregulated levels of cytokines detected on this group could be also attributed to the age of the patients, as the immune system response changes with age. Please discuss how the age of your patient's cohort can impact on your results.
On the other hand, the correlation of each cytokine with the clinical parameters GI, PD and CAL could be shown as correlation scatterplots. Thus showing the strenght of the correlation in a visualized way.
Minor Comments: (typos corrections)
1) Line 146: inchronic (separate)
2) Table 2: Meandifferenceofthelevelsof (separate)
3) Table 3: Group B- 0.4570.19 (separate)
4) Table 3: Group C- 0.6620.13 (separate)
5) Line 181: c (erase)
6) Line 181: chemokinescauses (separate)
7) Line 198: dysbioticmicrobiome (separate)
8) Line 227: replete (rephrase)
9) Line 233: animportant (separate)
10) Line 236: IL 21 (add line)
11) Line 248: IL17 (add line)
12) Line 290: ILsin (separate).
Author Response
|
Sr. no. |
remark |
Response to remark |
|
Reviewer 2
|
Major Comments: · I would suggest that on Table 1 you only show age, sex (please add this parameter) and clinical parameters (PI/GI/PD, etc.) evaluated for each group. Then, the results obtained for the cytokines concentration for each group (main finding of the study) may be presented on a separate figure including the p values shown on table 2. The p values obtained by tukey´s posthoc test can be incorporated into the figure and/or showed on as a supplementary table, as they are not a result itself, but a statistical analysis. · Along the cohort of patients, group C has the highest mean age. I understand that chronic periodontitis is more prevalent on older people, but the upregulated levels of cytokines detected on this group could be also attributed to the age of the patients, as the immune system response changes with age. Please discuss how the age of your patient's cohort can impact on your results.
· On the other hand, the correlation of each cytokine with the clinical parameters GI, PD and CAL could be shown as correlation scatterplots. Thus showing the strenght of the correlation in a visualized way.
· Minor Comments: (typos corrections)
|
Necessary modifications made, as suggested and relevant details have been added in the manuscript.
Tables have been reformatted adhering to the suggestions by the esteemed reviewers, with best of our knowledge. Table 1 in the revised version depicts only show age, sex ( details added, as suggested ) and clinical parameters (PI/GI/PD, etc.) evaluated for each group. Table 2 shows the intergroup comparison of the mean difference in the levels of interleukins using a posthoc Tukey’s test along with the mean and standard deviations of each interleukin.
The effect of age , as pointed and suggested by the reviewers have been mentioned and discussed with 2 new relevant citations .
Clark D, Kotronia E, Ramsay SE. Frailty, Aging, and Periodontal Disease: Basic Biological Considerations. Periodontol 2000 2021; 87(1): 143–56.
Milan-Mattos JC, Anibal FF, Perseguini NM, Minatel V, Rehder-Santos P, Castro CA, Vasilceac FA, Mattiello SM, Faccioli LH, Catai AM. Effects of natural aging and gender on pro-inflammatory markers. Braz J Med Biol Res. 2019;52(9):e8392. doi: 10.1590/1414-431X20198392. Epub 2019 Aug 12. PMID: 31411315; PMCID: PMC6694726.
Sorry, We could not provide with the scatter plots as unfortunately, we had lost the raw data of the individual patient, but only had the statistical analysis, we got conducted at the statistical department We have rearranged the correlation data in a single table as Table 3 for a concise and clear presentation.
Necessary modifications made, as suggested throughout the manuscript |

Reviewer 3 Report
The authors have reported a cross-sectional study where they have investigated the GCF levels of IL-17, IL-18, and IL-21 among a group of healthy, gingivitis, and periodontitis patients.
This study only gives baseline values but would have added more meaning if the randomized controlled trial were done and the interleukin levels were post-intervention.
There is no novelty in this study, as there is plenty of literature present.
Abstract:
The author uses the term novel cytokines. Although this term might not be suitable for this situation. There is a plethora of literature assessing interleukin 17, 18, and 21.
Line 30: “ After proper history taking” what does proper history taking mean. Rephrase.
Line 36: it would be best to include the mean and SD of the interleukin levels for each group to compare the differences.
In the introduction, the authors should cite the existing literature and indicate how this study is different from the existing literature.
Methods:
Line 98: Why was the age range 20 years to 50 years selected.
Line 99: what type of medication was considered an exclusion
Line 103: What is the ethics number
Line 104: Was sample size calculation/ power calculation done?
Line 115: For group C, the definition for periodontal disease, was this used previously by other authors.
The newer classification system classifies periodontal disease according to staging and grading, it is recommended to use that and not use terms such as Chronic periodontitis as it can lead to confusion.
The authors mentioned PD≥ 5m is periodontitis and ≤3mm is gingivitis. Then what about 4mm.
How was probing depth, and clinical attachment level measured? What kind of instruments were used? Did the authors’ record do a full mouth examination? How many sites were examined?
For the ELISA test, the process needs to be explained in brief. Did they check for control blanks?
The tables are not formatted properly and there are a lot of typos.
Why did the authors use micropipette instead of period paper and Periotron. How much was the GCF flow rate?
Author Response
|
Sr. no. |
remark |
Response to remark |
|
Reviewer 3
|
The authors have reported a cross-sectional study where they have investigated the GCF levels of IL-17, IL-18, and IL-21 among a group of healthy, gingivitis, and periodontitis patients. This study only gives baseline values but would have added more meaning if the randomized controlled trial were done and the interleukin levels were post-intervention. · There is no novelty in this study, as there is plenty of literature present. Abstract: The author uses the term novel cytokines. Although this term might not be suitable for this situation. There is a plethora of literature assessing interleukin 17, 18, and 21. · Line 30: “After proper history taking” what does proper history taking mean. Rephrase.
· Line 36: it would be best to include the mean and SD of the interleukin levels for each group to compare the differences. · In the introduction, the authors should cite the existing literature and indicate how this study is different from the existing literature.
Methods: · Line 98: Why was the age range 20 years to 50 years selected. · Line 99: what type of medication was considered an exclusion
· Line 103: What is the ethics number
· Line 104: Was sample size calculation/ power calculation done?
· Line 115: For group C, the definition for periodontal disease, was this used previously by other authors. · The newer classification system classifies periodontal disease according to staging and grading, it is recommended to use that and not use terms such as Chronic periodontitis as it can lead to confusion.
· The authors mentioned PD≥ 5m is periodontitis and ≤3mm is gingivitis. Then what about 4mm.
· How was probing depth, and clinical attachment level measured? What kind of instruments were used? Did the authors’ record do a full mouth examination? How many sites were examined? · For the ELISA test, the process needs to be explained in brief. Did they check for control blanks?
· The tables are not formatted properly and there are a lot of typos.
· Why did the authors use micropipette instead of period paper and Periotron . How much was the GCF flow rate?
|
We are in complete agreement to your valuable remark. In fact, the authors also on the same note, have suggested for future RCTs in the conclusion of the manuscript. However, the current investigation has been conducted as a preliminary study to explore the level of cytokines- IL-17,18 and 21 in GCF of healthy, gingivitis and periodontitis.
The term ”novel” has been removed, as suggested.
The sentence has been rephrased ,as suggested and the term ”proper” has been replaced by “complete” to clearly convey the intended meaning of the sentence..
Necessary modifications made, as suggested and relevant details have been added.
Necessary modifications made, as suggested and relevant details have been added in the “introduction” section (line no. ) along with the addition of new references in the bibliography.
Relevant details mentioned, as suggested. (The subjects included in the study were of the age group 20-50 years (most common age group visiting the hospital and to avoid old age related parameter variations), non-smokers, free from any known systemic disease and had not undergone any periodontal therapy or had received any antibiotics and anti‑inflammatory drugs in the previous six months. Pregnant and lactating females were also excluded.)
(RADC/Perio /28/2018, dated 30/10/2018).This has been duly mentioned in the revised manuscript.
Necessary modifications made, as suggested and relevant details have been added in the “Materials and methods;” section (line no. )
Yes, this has been the standard criteria of classification (AAP,1999) and has been used by numerous authors previously. Since at the time of the conduct of current work, no such concrete categorization in context of biomarkers is there and the new system of classification is yet being adopted in the clinical settings worldwide, esp in South Asia, thus the current work has been carried out on the basis of previously existing nomenclature of the periodontal disease.
Sincere apologies for the inadvertent typing error, which has been rectified in the revised version.
Necessary modifications made, as suggested and relevant details have been added in the “Materials and methods;” section (line no. )
Necessary modifications made, as suggested and relevant details have been added in the “Materials and methods;” section (line no. )
Necessary corrections made.
The use of microcapillary for the current study was opted owing to its distinct advantages mentioned above, in addition to being a much convenient and cost effective method for collection of GCF in our research setting. The microcapillary pipette was gently placed at the entrance of the gingival crevice and then a fixed volume of the sample was collected, irrespective of the flow rate of the GCF. |

Round 2
Reviewer 1 Report
The manuscript has been correctly revised, it can be published
Reviewer 3 Report
No further correction is required.